# Health system response to COVID-19 among primary health care units in Ethiopia: A qualitative study

Getnet Mitike[1]*, Frehiwot Nigatu[1], Eskinder Wolka[1], Atkure Defar[2], Masresha Tessema[2], Tezita Nigussie[1]

1 International Institute for Primary Health Care-Ethiopia, Addis Ababa, Ethiopia, 2 Ethiopian Public Health Institute, Addis Ababa, Ethiopia

* getnet.mitike@iphce.org

## Abstract

### Introduction

There was limited data on the experiences and roles of sub-national health systems in the response against COVID-19 in Ethiopia. This study explored how sub-national primary health care units and coordinating bodies in Ethiopia responded to COVID-19 during the first 6 months of pandemic.

### Methods

We conducted a qualitative study with descriptive phenomenological design using 59 key informants that were purposively selected. The interviews included leaders across Ethiopia's 10 regions and 2 administrative cities. Data were collected using a semi-structured interview guide that was translated into a local language. The interviews were conducted in person or by phone. Coding and categorizing led to the development of themes and sub-themes. Data were analyzed using thematic analysis.

### Results

Local administrators across different levels took the lead in responding to COVID-19 by organizing multisectoral planning and monitoring committees at regional, zonal and woreda (district) levels. Health leaders reacted to the demand for an expanded workforce by reassigning health professionals to COVID-19 surveillance and case management activities, adding COVID-19-related responsibilities to their workloads, temporarily blocking leave, and hiring new staff on contractual basis. Training was prioritized for: rapid response teams, laboratory technicians, healthcare providers assigned to treatment centers where care was provided for patients with COVID-19, and health extension workers. COVID-19 supplies and equipment, particularly personal protective equipment, were difficult to obtain at the beginning of the pandemic. Health officials used a variety of means to equip and protect staff, but the quantity fell short of their needs. Local health structures used broadcast media, print materials, and house-to-house education to raise community awareness about COVID-19. Rapid response teams took the lead in case investigation, contact tracing, and

**Data Availability Statement:** All relevant data are within the paper and Supporting Information files.

**Funding:** The author(s) received no specific funding for this work.

**Competing interests:** The authors have declared that no competing interests exist.

sample collection. The care for mild cases was shifted to home-based isolation as the number of infections increased and space became limited. However, essential health services were neglected at the beginning of the pandemic while the intensity of local multisectoral response (sectoral engagement) declined as the pandemic progressed.

## Conclusions

Local government authorities and health systems across Ethiopia waged an early response to the pandemic, drawing on multisectoral support and directing human, material, and financial resources toward the effort. But, the intensity of the multisectoral response waned and essential services began suffering as the pandemic progressed. There is a need to learn from the pandemic and invest in the basics of the health system–health workers, supplies, equipment, and infrastructure–as well as coordination of interventions.

## Introduction

After the World Health Organization (WHO) declared Public Health Emergency of International Concern in Early January 2020, COVID-19 became a global threat and a pandemic in March 2020 [1]. The COVID pandemic had so far affected and killed millions of people globally. By the end of 2020 (December 28), there had been a total 4,162, 931 confirmed cases and 81,404 deaths [2]. From the beginning, COVID-19 caught the attention of governments for fear of its multifaceted impacts on populations, health systems, economies, social relationships, and political repercussions. In Africa, early projections and learning from the impacts of Ebola outbreak, expected calamities from COVID-19 was alarming, This was compounded by the fragile health systems, weak health security capacity and poor economies [3–8].

In Ethiopia, the first case of COVID-19 was detected on March 13, 2020. In April 2020, the government issued a five-month state of emergency that closed schools and universities, banned gatherings of more than four people, ordered transportation service providers to reduce their passenger loads, prohibited sports activities, and mandated the wearing of masks in public [9,10]. In collaboration with the WHO and other partners, the government also began assessing and equipping public and private laboratories to expand testing for COVID-19 [11].

Much of the early published research work in Africa particularly from Ethiopia that examined national responses were based on document reviews [12–21], and the empirical studies, focused on knowledge, attitude and practices toward COVID-19 prevention measures among community members and health professionals [22–35]. Furthermore, approaches and processes to systematically mount the response against COVID-19 were largely unknown particularly at subnational (regional) level where most of the preventive activities are taking place. So, exploring the response including organizational approaches towards COVID-19 at subnational and primary health care (PHC) levels in a decentralized system such as Ethiopia will offer a better picture of the process, measures undertaken to control and prevent the spread of COVID-19. This study was therefore to understand the process and learn from the response towards COVID-19 pandemic at subnational level in Ethiopia considering the first 6 months of the pandemic.

## Materials and methods

### Study setting

The study was conducted in September and October of 2020 in Ethiopia's ten regions (Afar, Amhara, Benishangul-Gumuz, Gambella, Harari, Oromia, Sidama, Somali, Southern Nations

Nationalities and Peoples, and Tigray) and two administrative cities (Addis Ababa and Dire Dawa). The regions and administrative cities are divided into zones or sub cities that are further divided into woredas (districts), and each woreda consists of kebeles, which are the lowest administrative units. The health system is organized into three tier system represented by primary health care units (primary level), general hospital (secondary level) and tertiary referral hospital (tertiary level) each with catchment populations. A Primary Health Care Unit (PHCU) consists of a health center, five health posts and a primary hospital [36]. The PHCU and corresponding decentralized administrative structures are shown in S1 Fig.

## Study design

We employed descriptive qualitative (phenomenological) design to explore the Ethiopian health system's response to COVID-19 in the first six months of the pandemic.

## Selection of participants and sample size

We used purposive sampling to select a total of 59 participants. Key informants were chosen among health professionals who were actively engaged in the COVID-19 response across the country. Active engagement included direct participation in decision making pertaining to prevention and control and provision of services to patients affected by COVID-19. From each region and city administration, we selected one person from a regional health bureau, two persons from zonal/woreda health offices, and two from health facilities. The criteria for selection were completion of post-secondary education, current employment in the public health system and involvement in COVID-19 response. In our sampling strategy we covered all regions (subnational) of the country and followed the decentralization system with the aim of obtaining some degree of representation of the administrative and service provision capacities at various levels. This was important as there were differences in the capacities of the regions with regards to human resource, distribution of facilities and modalities of health services provision.

## Data collection and analysis

We recruited and trained 12 data collectors (interviewers) and 12 supervisors. Both groups had graduate level education with some qualitative research experience. Because of government restrictions on assembly and travel, we held training for data collectors and supervisors remotely via Google Meet for half a day. The data collectors conducted interviews in person or by phone using a semi-structured interview guide that was translated into Amharic, the national language. The data collectors digitally recorded the interviews and translated and transcribed the responses into English. The supervisors met (in person or via phone) with the data collectors each day to review the transcripts and discuss ways to improve the quality of data collection. The supervisors provided feedback and support in consultation with the PI.

Two experts with previous experience and training in qualitative methods read the transcripts independently. The PI is trained in qualitative methods at graduate level and has co-authored papers in peer reviewed journals and the other co-investigator has graduate level training in social work and public health. The transcripts were coded and a codebook was developed. The codes were revised several times over the course of the analysis between the PI and the co-investigator. From the codes themes and sub themes were developed.

Validity and reliability of qualitative research is measured through trustworthiness which in turn includes credibility, confirmability, transferability and dependability [37]. To ensure the credibility of our findings, we recruited data collectors and supervisors who were actively working in the respective regions and who were familiar with their respective study areas and health systems. The data collectors and supervisors also received training on the purpose,

methods, and tools of the study. We accounted for confirmability by capturing participants' responses through recordings and transcriptions that were checked by supervisors and read several times by investigators. The study addressed transferability by including all regions of the country as the infrastructures and context are at varying levels. We also validated the semi-structured guides through discussion with the interviewers and supervisors. For ensuring dependability, the guides for the interview included interventions and strategies recommended by the WHO and the country context. The methods and procedures applied were consistent with other authors that conducted similar studies.

## Ethical consideration

The proposal was approved by the Institutional Review Board of Ethiopian Public Health Institute (EPHI) (# 280–2020). All study participants provided informed verbal consent to be interviewed. Verbal consent was used to minimize contact through paper exchange considering COVID-19 transmission.

## Results

### Study Participants

This qualitative analysis was conducted using the transcripts of fifty-nine (59) health care professionals that participated in the study. This was less than the expected (60). In one of the regions only four out of five expected participants were available for the interviews. Eighty-nine percent of the participants were male. Participants ranged substantially in age, with a mean age of 35 years. Forty four percent of the health facilities were well represented in the sample. Participants were in senior leadership or management positions, but, on average, had been on the job less than five years (Table 1).

The results are categorized into 5 themes and subthemes: 1) Early Activities: Containment, Coordination, and Incidence Management; 2) Operational Support and Logistics (have three sub themes; *Health workforce*, *Training*, and *COVID-19 supplies and equipment*); 3) Risk Communication and Community Engagement; 4) Surveillance, Rapid Response Teams, and Case Investigation; 5) Case Management and Infection Prevention and Control (two sub themes: *Case management and risk infection prevention and control*).

### 1. Early activities

**Containment, coordination and incidence management.** Preparations for containment measures were implemented in Ethiopia in regions that share borders with neighboring countries and in two cities with international airports. This was before March 13, 2020 when the first cases of SARS-COV-2 were reported. Some of measures included screening at points of entry such as airports and cross border sites (two-third of regional and city administrations). Accordingly, regional and zonal leaders instituted new entrant's screenings at the points of entry. For example, regional leaders in Afar established entrance screening at a major border crossing with Djibouti and airport screening was set up in Addis Ababa. These activities were instituted based on the WHO declaration and the alert letter from the Ethiopian Public Health Institute. Containment measures were based on the guidelines and case definition of the WHO which was communicated to the subregional levels.

After the first cases of COVID-19 were confirmed, the federal government established coordination mechanisms that cascaded at subnational levels i.e., 1) National Disaster Risk Management Council led by Deputy Prime Minister's Office 2) Public Health Emergency Management (PHEM) led by the Ministry of Health 3) The PHEM Technical Taskforce led by

**Table 1. Characteristics of study participants in selected primary health care units, Ethiopia, 2020\*.**

| Characteristics | Number (percent) |
|---|---|
| **Age in years** *(n = 58)* | |
| Mean | 35 |
| Range | 26–58 |
| **Sex** *(n = 59)* | |
| Male | 53 (89.8%) |
| Female | 6 (10.2%) |
| **Highest level of education** *(n = 59)* | |
| BSc | 28 (47.5%) |
| MA/MSc/MPH | 25 (42.4%) |
| MD | 4 (6.8%) |
| Other | 2 (3.4%) |
| **Profession** *(n = 59)* | |
| Public health | 26 (44.1%) |
| Nursing | 10 (16.9%) |
| Health Officer | 10 (16.9%) |
| Medicine | 4 (6.8%) |
| Environmental health | 3 (5.1%) |
| Midwifery | 2 (3.4%) |
| Other | 4 (6.8%) |
| **Level** *(n = 59)* | |
| Health facility | 26 (44.1%) |
| Regional health bureau | 16 (27.1%) |
| Zonal health department | 8 (13.6%) |
| Woreda health office | 8 (13.6%) |
| Unknown | 1 (1.7%) |
| **Position** *(n = 58)* | |
| Head of zonal health department, woreda health office or health facility | 26 (44.8%) |
| Head of division, team, or service | 16 (27.6%) |
| Public health emergency manager | 7 (12.1%) |
| Deputy head of regional health bureau | 5 (8.6%) |
| Other | 4 (6.9%) |
| **Number of years in position** *(n = 58)* | |
| Mean | 4 |
| Range | 2 months to 14 years |
| **Total years of professional experience** *(n = 59)* | |
| Mean | 12 |
| Range | 2–40 |

\* The *n* varies for some questions due to non-response. Overall, N = 59.

the Director-General of the Ethiopian Public Health Institute, and 4) the PHEM Technical working Group led by the national Incident Manager [38]. Following this, various structures had been set at the regions with varying constituencies of coordinating committees and leadership arrangements [38] including at zonal, woreda and facility levels. The administrative bodies–from regional councils to lowest administrative units (kebeles)–formed coordination committees. The heads of administrative bodies presided over these coordinating bodies (multisectoral committees) that planned, organized, and monitored local responses. Different

sectors were represented on these committees, including health; education; finance; law enforcement; water, irrigation and electricity; and trade. These committees were called task-forces and were established all-over the regions depending on their sector arrangements. The taskforces met regularly and in some cases they met every three days.

Sub-committees or technical committees were also formed to implement specific tasks. Key informants frequently cited the finance or resource mobilization sub-committee, which was responsible for obtaining support—in cash and kind—from local governments, non-governmental organizations, and community members. The coordination committees also raised and directed resources to the most vulnerable members of their communities including people living with daily subsistence, long distance drivers, police stations, and street dwellers. For instance, one of the respondents replied, *"Another committee established by the mayor was the resource mobilization committee, which includes the finance and health sectors, and coordinates resource mobilization to purchase personal protective equipment like facemasks and provides support for community members who are seriously affected."* (Sidama, Zonal Health Department).

Surveillance was a key focus of incidence management teams across regions, particularly at the woreda level. At higher levels, the initial focus was on operations: assessing the extent of need for and setting up quarantine, isolation, and treatment centers; customizing or developing guidelines for health workers; and expanding screenings at major entry points. Health centers organized rapid response teams to conduct case finding and contact tracing in their communities.

*"Before health centers began any activity, they organized committees. The committees consisted of seven or eight people, including health professionals and supportive staff. After establishing the committees, we divided tasks such as providing community awareness and availing supplies. . . . if you look at the report there were committees both in 2012 and 2013 [Ethiopian calendar] and those committees had sub-committees that provided support for those in need at the community level, which included distributing sanitizers, facemasks, food, and money. In addition, there were also committees working on providing orientation and education for health professionals."* (Oromia, Health Facility 1).

## 2. Operational support and logistics

**Health workforce.** As most key informants described, health leaders responded to the heightened demand for healthcare workers in two ways: 1) by redeploying or adding responsibilities to the existing workforce, and 2) by hiring new workers. Often, authorities adopted the first approach. Health professionals from hospitals and health centers were redeployed to isolation and treatment centers, rapid response teams (RRT), and care teams for patients in home-based isolation. They were also asked to serve in woreda health offices and regional health bureaus as members of planning and coordination teams.

Health extension workers, who were already engaged in community education, mobilization and prevention activities, added COVID-19 related tasks to their day-to-day activities by leading community-based education, screening efforts and assisting RRPs with contact tracing. The added responsibilities created pushback: "*Most services, including community services and COVID, rest on the shoulders of the health extension workers. That is why they complained about the work burden.*" (Sidama Health Facility 2).

To maintain the capacity of the existing workforce, the health system instituted a 'no leave policy' that denied health workers their annual leave. "*Annual leave was prohibited to all professionals because if all professionals were not working together, the pandemic could affect many people.*" (Harari, Regional Health Bureau). However, during the state of emergency, certain

categories of workers–pregnant and lactating women and providers with chronic conditions–were ordered to stay home.

Additionally, health authorities in Addis Ababa, Afar, Benishangul-Gumuz, Oromia, Southern Nations, Nationalities and Peoples' (SNNP), and Tigray regions hired new staff. They received money from the regional health bureau, regional government, or Ministry of Health to increase their health workforce, but often temporarily. The new hires were largely professionals—medical doctors, health officers, environmental health experts, laboratory technicians, and nurses. Despite the investment in health professionals, key informants in Addis Ababa complained of a severe shortage of urban health extension workers in the capital.

**Training.** Training for the health workers focused on key pillars of COVID-19 response —surveillance (specifically, case investigation by RRTs), infection prevention and control, community education, and case management. Health authorities relied on the train-the-trainer model to transfer skills and information from regional health bureaus down to health centers and health posts. More often, however, regional health authorities suspended the linear train-the-trainer model and directly offered training to staff at lower levels.

Content specific trainings were organized for RRTs, staff working at quarantine, isolation and treatment centers, laboratory technicians, and health extension workers. In Amhara, informants mentioned that influential community members and health professionals from the private sector were also invited to participate in trainings, particularly on infection prevention and control (IPC). The duration of trainings differed based on the topic. Trainings were held online, in venues that were large enough to meet physical distancing requirements, or on-site.

In most regions, trainings organized by regional or zonal offices were supported by partner institutions and local universities. In addition, WHO, EPHI and MOH provided technical support.

## COVID-19 supplies and equipment

Health officials and providers across the country noted a shortage of COVID-19 supplies and equipment, particularly PPE, at the beginning of the pandemic. For some, access to PPE remained a problem as late as September/October 2020. Health officials reported that, initially, they faced a shortage of facemasks and protective gowns, and to a lesser extent, gloves. In addition, they lacked the beds, linens, and medical devices needed to furnish isolation and treatment centers. Hand sanitizers and cleaning materials were also in short supply. A zonal health official in Tigray spoke about their struggle to obtain reagents for laboratories. Problems accessing non-PPE materials, however, were not uniform: materials that were scarce in one region were available in others. For example, a zonal health official in Addis Ababa stated "*There were no problems with sanitizers*." While, according to a health center director in Harari, "*There was a greater shortage of sanitizers than gloves.*"

In the early months of the pandemic, the challenge in purchasing COVID-19 supplies and equipment, particularly PPE, was largely due to mismatch in supply and demand. The demand for facemasks, gloves, and sanitizers outpaced their production. Even when health centers or offices had the budget to purchase PPE, the items were not available in the market. "*At the beginning it was challenging, there was almost no PPE. There was no production of PPE and supply. Even though we tried to buy from the market, it was not available. Later when we invested in PPE—sanitizers, masks and other items—it was challenging financially. It affected the other services significantly.*" (Tigray, Health Facility 1).

Health officials used different strategies to procure PPE and other needed materials. Some repurposed materials that had been in storage for public health emergencies such as Ebola and cholera. Health centers in particular used a combination of their own revenues, shifting

budgets to COVID-19 and through assistance from regional health bureaus, city administrations and private donors to get needed supplies. The budget shafting from other services to COVID-19 was illustrated in the following example. "...we *used finance allocated for non-COVID services to buy PPEs and other resources, which ultimately affected the essential health services...*". (Addis Ababa, Health Bureau).

The private sector played a role in alleviating demand-associated shortfalls for COVID-19 supplies and equipment. Businesses, non-governmental organizations, and individuals, both in Ethiopia and abroad, were mobilized to support the health system in this effort. A zonal health leader in Amhara indicated that international non-governmental organizations (NGOs) were instrumental in procuring supplies–like alcohol and sanitizers–that were difficult for health authorities to purchase from the market. Yet, while NGOs and individuals made donations, businesses in a few regions—ranging from microenterprises in Tigray to an industrial park in Sidama—manufactured needed supplies.

In a few regions—Harari, Addis Ababa, Afar and Amhara–study participants across different tiers of the health system reported that their supply issues had been resolved after once they were made available in the market. Most of the health facilities received support from a city administration, and health centers in Afar and Addis Ababa supplemented this support by using internal revenues.

Despite their efforts to obtain needed materials, most study participants shared that the PPE items available were inadequate either in quality or, more often, in number. To conserve PPE, health officials resorted to rationing or recycling. At a health center in Addis Ababa, each health professional received 50 masks that were expected to last for 90 days. Health center officials in Oromia requested that health workers wash and reuse masks that were intended to be worn for only four hours. A regional health bureau official from Amhara spoke about prioritizing PPE for professionals who worked in laboratories, isolation and treatment centers "*even though all health workers were demanding all PPE...*" In the end, an inadequate supply of PPE made it difficult to follow guidelines meant to protect health care providers and clients, which in turn, heightened fear of infection from all sides.

## 3. Risk Communication and community engagement

Risk communication and community engagement was central to the response effort from the beginning of the pandemic. At regional and zonal levels, risk communication teams in incident management offices developed and disseminated messages with support from partner organizations and local government communications offices. Printed materials, radio, and television were commonly used to transmit information, with higher-level offices relying on broadcast media. Regional and zonal offices ran ads on local radio and television stations about the pandemic. Experts appeared on live television or radio programs to answer audience questions or present the latest information about the virus. As the following quote attests, collaborations with international NGOs, academic institutions, and professional associations helped regional and zonal leaders to maximize their communication efforts: "*We also worked with the Amhara Physicians' Association, clinicians, and public health experts who made live appearances on television programs in collaboration with Bahir Dar University.*" (Amhara, Regional Health Bureau).

To a lesser extent, regional and zonal health offices also used social media, particularly Facebook, to share information and updates. The regional health bureau in Tigray, for instance, posted the daily number of confirmed cases in the region on its website and Facebook page. Visitors to these sites could also find information about COVID-19 and steps local authorities were taking to contain the pandemic. In addition to traditional and social media, health officials in Sidama and SNNP collaborated with artists to develop music and drama about

COVID-19: "*Artists in the Sidama region communicated messages through songs and drama, for adults and children, on how to prevent the disease.*" (Sidama, Regional Health Bureau). A woreda health official from SNNP spoke of a "holistic" song produced by senior Wolaita musicians that gained popularity with rural communities. In Oromia, community and religious leaders played a role in community education. "*We provided information to the community through religious leaders, cultural leaders—Aba Geda, and different media. People in our region have received sufficient information about prevention and have used that information to keep themselves safe.*" (Oromia, Regional Health Bureau).

Woreda health offices and health centers commonly relied on printed materials (leaflets, brochures, posters, and banners) to spread information about the COVID-19. Regional or zonal offices created these materials in local languages and sent them to the lower tiers for distribution. Health extension workers (HEWs), volunteers, and care providers handed out brochures and fliers in the community and during house-to-house visits. Posters and banners were displayed in crowded markets and busy streets.

With decentralized coordination mechanism in place (federal, regional, zonal, woreda and kebele levels), intensive outreach took place at different sites—homes, work places, places of worship, and health facilities. Health professionals, teachers, and volunteers visited households in Tigray and in Somali regions and also health extension workers, traditional birth attendants, and family health teams visited households. Health professionals also used loud speakers, often driving through city streets or villages, to maximize their reach. Health centers and woreda health offices made a deliberate effort to target vulnerable communities. For example, a health center in Amhara conducted outreach to the homeless and a woreda in Addis Ababa shaped its message about preventive measures to address the concerns of community members with non-communicable diseases (NCDs) as they are at high risk for infection. Taskforces and technical committees at woreda/health center levels coordinated all the activities with the help of corresponding law enforcements.

Risk communication messages stressed the importance of maintaining hand hygiene (using soap and water or hand sanitizers), wearing masks, keeping a physical distance. Messages also covered the signs and symptoms of the disease and provided hotline numbers to call for additional information or to report suspected cases. During household visits, health extension workers demonstrated proper hand washing, physical distancing, and mask wearing techniques. "*HEWs did house-to-house visits in their assigned areas or kebeles. They supplied households with soaps and businesses with hand sanitizers. They taught households how to prevent COVID-19, advised them to stay home, and to wear masks if they were in public. They also gave out leaflets with the emergency telephone number.*" (Somali, Health Facility 2).

A health center in Harari encouraged community members with chronic conditions to continue taking their medications. Still, financial, political, and environmental pressures placed limits on risk communication efforts in some regions. A representative from the Addis Ababa regional health bureau said "*the rate of mass media utilization is not as expected due to the expense.*" Also, in Sidama, a woreda health official noted that "*performance was very strong initially, but it has become weak recently because of cost and our lack of capacity*" in reference to the use of mobile public announcement systems to disseminate information about the virus. Outreach was stymied in rural parts of Harari due to security issues and heavy rainfall that made roads inaccessible.

## 4. Surveillance, rapid response teams, and case investigation

Health officials relied on syndromic screening to isolate possible cases of COVID-19. Screenings were based on a case definition published by the Federal Ministry of Health. The most recent definition was published in May 2020 and a health center director in Addis Ababa

suggested that even prior to this, modifications were being made to the definition at lower levels [39]. Study participants also highlighted the limits to pointed syndromic-based case definitions in the context of high rates of asymptomatic infections.

Health extension workers conducted house-to-house screenings in three rounds with a testing campaign that occurred in August 2020. Community screenings targeted high-risk groups and places where transmissions were most likely to occur: long-distance bus drivers for instance, were prioritized for screening in Amhara and Tigray and screenings took place at bus station and police stations in Gambella. Community members were also encouraged to contact health authorities if anyone in the community showed signs or symptoms of the disease. Until 30 September 2020, a total of 1,272, 352 samples were tested out of which 75,368 were confirmed cases of COVID-19.

Health centers set up triage areas to screen every patient who entered their compounds. Screeners used a checklist to identify suspected cases. The initial screening consisted of a temperature check and questions probing signs and symptoms of acute respiratory infection. Patients with fever and cough were escorted to an isolation room in the health center where they were further evaluated by a clinician. Patients who were suspected of COVID-19 remained in the isolation room until they were tested.

As the number of cases that required hospitalization grew, health authorities expanded the number of treatment-centers, quarantine and isolation centers in each region–often selecting schools and universities (which were closed at the time (2020) after the state of emergency was declared) as supplementary sites and hotels provided by local business owners.

Zonal health officials in Sidama opted to use space donated by local businesses for COVID-19 treatment rather than interrupt the delivery of essential services at health facilities. A regional health bureau official in Afar reported that, "*with the surge of infections, severe cases of COVID-19 were placed in hospitals and milder cases were treated in non-traditional care facilities*". Many health centers set up isolation areas by converting a room or unused area in their facilities. In some centers, the conversion of spaces encroached on existing services, as in the case of health centers in Benishangul-Gumuz where waiting rooms for expectant mothers were transformed into isolation rooms for suspected COVID-19 cases. Isolation rooms were outfitted with beds, oxygen cylinders, and pulse oximeters in Addis Ababa and Tigray.

The RRT was called to conduct an investigation. when there was a suspicious finding in the community, The team—which typically consisted of a clinician, an epidemiologist, an environmental health expert, a health educator, and a lab technician–then transported individuals suspected of COVID-19 to a quarantine center where they were tested and remained until their results were known.

Depending on the region, either staff from the EPHI or members of RRTs were responsible for sample collection. In August, the federal government undertook a nation-wide campaign to gauge the level of infection in the country and measure community understanding of COVID-19 prevention measures. Vulnerable groups, such as those with chronic conditions, were specifically targeted for testing during the campaign.

The RRTs were also responsible for transporting samples to the nearest laboratory. Each region had at least one laboratory and many were located on the campuses of major universities. Before the regional laboratories were set up, tests were sent to laboratories in Addis Ababa or outside the country. Key informants in Oromia and Tigray noted that laboratories struggled to report test results in a timely manner. Delays were reportedly due to a lack of coordination, limited diagnostic machines, and a lack of reagents. *"Quarantine is located at another site. . .. but the problem is test results are not reported timely and a lot of complains are coming from people staying at the quarantine center. Results are being reported 9 or 10 days after they are expected, which is extremely frustrating. . . ."* (Tigray, Woreda Health Office).

### 5. Case management and infection prevention and control

**Case management.**  Individuals in quarantine were released after seven days if they tested negative. Otherwise, they were transferred to an isolation or treatment center. Rapid response teams conducted contact tracing and suspected contacts were placed in quarantine and tested. The policy changed to allow mild and asymptomatic cases to isolate at home as the state of emergency came to an end and schools prepared to re-open. Rapid response teams determined who could isolate at home and regularly followed-up.

**Infection prevention and control.**  Health centers undertook precautions to ensure the safety of their workers and patients. They set up washing stations where patients were required to wash their hands before proceeding to their appointments. They limited entry only to patients, placed markers two meters apart so that patients could adhere to physical distancing requirements, and disinfected essential departments. Some centers had two gates, one for entry and another for exit. *"In addition, guards were given additional work to make sure that every client entered to our health center washed their hands using water and soap available at the gate"*. (Oromia, Facility 2).

In regions across Ethiopia, community-based IPC was largely the responsibility of regional, zonal, and woreda health offices. Many installed handwashing facilities in crowded areas. A woreda in SNNP temporarily housed "street dwellers" who were considered high-risk for infection and provided them with food and hygiene materials. A zonal health department in Sidama disinfected sidewalks and office buildings every 15 days. Higher-level offices collaborated with the police to enforce or promote infection control measures in markets, restaurants, and passenger cars. They also organized volunteers, particularly youth and members of the women's development army (volunteer community health workers), to educate and encourage people to adhere to prevention strategies.

## Discussion

Similar to many other African countries, different regions of Ethiopia acted early by implementing screening measures at airports and border crossings before the first case of coronavirus was detected [40]. Following the confirmation of the first case, sub-national government structures from regional councils to kebeles set up multisectoral coordinating committees that were responsible for planning, organizing, and monitoring local responses. Research indicates that local governments also played a vital role in the pandemic response in other low- and middle-income countries. Similar to our findings, studies from China and India showed that local government efforts were multisectoral and focused on resource mobilization, infection control, and risk communication [41–43]. A county government outside Wuhan, China, for example, organized "functional groups" that, among other responsibilities, procured PPE from different sources for local public hospitals and community volunteers [42]. These studies also found out that local governments provided aid to needy individuals and households, enforced disease control measures such as physical distancing and screening and quarantine of travelers, and created public awareness [41,42]. These examples, as well as Ethiopia's experience has shown the potential for local governments to play a critical role in public health emergencies and other disasters. As Dutta and Fisher have argued, decentralized government institutions are better positioned to carry out the basic functions of disease control because they have a deeper understanding of community needs, can respond more efficiently to crises, and hold greater legitimacy with constituents than distant government officials [41]. Unfortunately, our study showed that the role of local governments was not sustained over the course of the pandemic–a finding that was not reflected in other studies. This might be due to limited experience in exercising decentralization in Ethiopia and lack of mechanisms to ensure continuous

financial support. Furthermore, trends in international and national practices could influence and this requires further investigation. Initially, the sub-national response in Ethiopia also embodied a whole-of-society approach that was promoted and reflected in the literature. This approach endorses collaboration across sectors based on the notion that no single body or institution can capably respond to complex emergencies or social problems [44]. Again, studies from India and China found that local governments mobilized community members to volunteer in the response effort [41,45,46]. Also, a survey of medical students, health workers, and community service workers in Vietnam showed strong support for involving youth groups, women's associations, religious leaders, and local organizations in community education, surveillance, and contact tracing during the pandemic which is similar to our study particularly at the initial phase [47]. In our study, participants highlighted and praised civic society institutions, volunteers, and businesses for engaging in a variety of activities, ranging from community education and the provision of supplies and facilities to the enforcement of infection control measures. This was a very good lesson. But, six months after the start of the pandemic, the study participants observed a decline in multisectoral engagement and warned of the health sector's capacity to control the pandemic in isolation.

Because of underinvestment in health care, many low-income countries suffer from a shortage of critical health inputs including, supplies, infrastructure, and health professionals. To fill the gap in human resources, for instance, experts advocate the use of community health workers in both maintaining essential health services and containing the pandemic [48,49]. This is viewed as a feasible solution in settings where community health workers outnumber clinicians and function as trusted sources of health information and services [50].

Our study showed that community health workers (known as health extension workers or professionals) in Ethiopia were a key to COVID-19 response strategies. They engaged in community education—principally through house-to-house visits—performed screenings, and assisted RRTs with contact tracing. A rapid evidence synthesis found that community health workers in low- and middle-income countries played similar roles in the pandemic response [51]. Furthermore, when essential health resources showed signs of decline in Ethiopia, community health workers were tasked with alleviating community members' fears of accessing health services and re-engaging patients who were lost to follow-up. In many areas, responsibilities related to COVID-19 were added to their existing workload, creating low morale and pushback from health extension workers as the time progressed. The rapid evidence synthesis showed that improving benefits, providing training and recognizing the contributions of community health workers in high-profile events could bolster their recruitment and retention [51].

In addition to mobilizing health extension workers, sub-national health systems in Ethiopia addressed the surge in the health workforce by reassigning health professionals, cancelling leaves of absence, recruiting new health workers, and bringing retired or private sector professionals into the workforce. While comparable research in low-income countries was not available, a study of 44 countries in Europe and North America showed that the governments of wealthy nations used mix of policy options similar to what we found out [52]. The most common strategies involved maintaining the existing workforce by extending work hours, moving staff from part-time to full-time, or suspending annual leaves. These governments also recruited medical and nursing students (a strategy that was not mentioned by participants in our study), redeployed workers to facilities or areas with the greatest need, and enlisted retired, foreign, and private sector professionals. Most countries used at least two strategies. While the impact of these measures in Europe and North America has not been studied, our study participants linked a weakening of the essential health services to the redeployment of staff to the COVID-19 response. This might be due to inadequate strategies and financial constraints in recruitment of new staff and redeployment of available health workers in Ethiopia.

An important factor in containing COVID-19 was protecting the health workforce from infection. Study participants shared their struggles to acquire PPE, particularly facemasks and gowns, at the beginning of the pandemic. Due to constraints in production, PPE was limited and costly. Ethiopia was not unique in this regard; countries worldwide faced a shortage of PPE and price hikes at the start of the pandemic [53]. But low-income countries were disproportionately affected because of their relatively weak bargaining positions in the global supply chain, low capacity in production of supplies, limited purchasing power, and low hands-on reserves of PPE [54–56].

When production began to accelerate, study participants accessed key protective gear through different means, including the engagement of local businesses. Local production of PPE was a solution to the shortage in a number of low-income countries: the governments of Uganda, Tanzania, Kenya, Nigeria, Liberia, and Ghana, for instance. [54–56]. At a more grassroots level, a government hospital in Dar es Salaam, Tanzania collaborated with seamstresses, business owners, community members, and NGOs to produce reusable facemasks, gowns, scrubs and caps for its staff [55]. However, reusable facemasks vary in quality and may not be the best option. Reusable masks vary from homemade cloth masks to masks with HEPA filter [57]. In Ethiopia, despite their efforts to meet the demand for PPE, most participants in our study were unsatisfied with the quantity of PPE at their disposal. Our finding is corroborated by a survey of health care workers at a public hospital in Addis Ababa in which over three-fourth of the respondents reported an inadequate supply of PPE at their hospital [58].

Risk communication and community engagement was central to the response effort in Ethiopia. Regional and zonal offices used broadcast media, and to a lesser extent social media such as Facebook, to disseminate information about the COVID-19. With support from international NGOs, academic institutions, and professional associations, they ran ads on local television and radio stations and invited experts to speak about the pandemic on broadcast media. However, risk communication efforts were limited in some areas by financial, political, and environmental obstacles, Most COVID-19 related studies conducted in Ethiopia examined knowledge, attitudes, and practices among community members, showing generally high rates of knowledge and poor prevention practices.

## Limitations of the study

Due to national and state level lockdowns the investigators could not go out to the field thus limiting the potential for extra lens to note field observations not captured by the interviews. However, the authors have tried to overcome this by assigning a supervisor for each interviewer in each region. We noted that data collected from two of the regions was not good enough in terms of depth and probes affecting the richness in some of the topics. This might be due to lack of experience of those interviewers in applying the different techniques of qualitative methods. However, the impact was minimal as the sample was large enough to explore the topics of interest.

## Conclusions

Sub-national health structures waged a swift and coordinated response to COVID-19 at the start of the pandemic. They drew on all available resources—from their own expertise and workforce to local governments and businesses, academic institutions, and community members to mitigate the spread of the virus. However, the response was strained from the beginning–due to a shortage of health workers, space, equipment and supplies–and began to fray as the pandemic worsened. The multisectoral response that started with strong political commitment and drove the initial response declined within six months of the pandemic. As the

number of infections in Ethiopia rise, we recommend: reengaging local administrations and actors outside the health sector, including academic institutions and local businesses, in the response effort; strengthening resource mobilization for COVID-19 supplies and equipment; improving testing capacity across the country; and collaborating with local media and community leaders to strengthen community adherence to prevention measures. There is a need to invest in the basics of the health system–health workers, supplies and equipment, and infrastructure–as well as specific interventions to intensify the COVID-19 response and restore the multi-faceted response. The study has illuminated relevant lessons for future emergencies and underscored engagement of local governments was a key to take immediate action and sustain the local response against COVID-19.

## Supporting information

**S1 Fig. PHCU and administrative structures.**
(DOCX)

## Acknowledgments

We would like to acknowledge the study participants and their respective regional health bureaus, zonal health departments, woreda health offices, and health centers for agreeing to take part in this study. We are also grateful to the data collectors and supervisors, staff at Ethiopian Public Health Institute and the International Institute for Primary Health Care-Ethiopia for their technical support.

## Author Contributions

**Conceptualization:** Getnet Mitike, Frehiwot Nigatu, Eskinder Wolka.

**Data curation:** Getnet Mitike, Eskinder Wolka, Tezita Nigussie.

**Formal analysis:** Getnet Mitike, Tezita Nigussie.

**Methodology:** Getnet Mitike, Eskinder Wolka, Atkure Defar, Masresha Tessema.

**Project administration:** Frehiwot Nigatu.

**Resources:** Frehiwot Nigatu.

**Software:** Getnet Mitike.

**Supervision:** Getnet Mitike, Frehiwot Nigatu, Eskinder Wolka.

**Validation:** Eskinder Wolka, Atkure Defar, Masresha Tessema, Tezita Nigussie.

**Writing – original draft:** Getnet Mitike, Tezita Nigussie.

**Writing – review & editing:** Getnet Mitike, Frehiwot Nigatu, Eskinder Wolka, Atkure Defar, Masresha Tessema, Tezita Nigussie.

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
