## [Decision Letter · Decision Letter 0]

3 Oct 2022

PONE-D-22-03390Health system response to COVID-19 among primary health care units in Ethiopia: A 

qualitative studyPLOS ONE

Dear Dr. Kassie,

Thank you for submitting your manuscript to PLOS ONE. After careful consideration, we feel that it has merit but does not fully meet PLOS ONE’s publication criteria as it currently stands. Therefore, we invite you to submit a revised version of the manuscript that addresses the points raised during the review process.

We look forward to receiving your revised manuscript.

Kind regards,

Hubert Amu

Academic Editor

PLOS ONE

Journal Requirements:

Reviewers' comments:

Reviewer's Responses to Questions

**Comments to the Author**

1. Is the manuscript technically sound, and do the data support the conclusions?

Reviewer #1: Partly

Reviewer #2: Yes

2. Has the statistical analysis been performed appropriately and rigorously? 

Reviewer #1: N/A

Reviewer #2: I Don't Know

3. Have the authors made all data underlying the findings in their manuscript fully available?

Reviewer #1: No

Reviewer #2: Yes

4. Is the manuscript presented in an intelligible fashion and written in standard English?

Reviewer #1: No

Reviewer #2: No

5. Review Comments to the Author

Reviewer #1: The study, 'Health system response to COVID-19 among primary health care units in Ethiopia: A qualitative study', is a qualitative study as stated. It explored the epidemic situation of COVID-19 in Ethiopia basically from the response and management points of view. This is a necessary study but for a qualitative study, the write up and description of the results must be logically presented in clear unambiguous manner to ensure easy flow and readability. The authors may relook into how the result areas are written. It was somehow long and tortuous.

Specific comments to improve the manuscript and queries have also been raised within the body of the manuscript which the authors may utilise to improve the document.

Reviewer #2: I want to ask about what about professionals and health workers who got infected, how you manage? What the effect of that on the health service? Also, why you do not involve assessment of modalities of online care that need extend rapid access to cellphones and internet for health professionals and users. Also, remote consultations with video, since it facilitates monitoring vital signs and decreased the risk of infection.

6. PLOS authors have the option to publish the peer review history of their article (what does this mean?). If published, this will include your full peer review and any attached files.

Reviewer #1: **Yes: **Folorunso Oludayo Fasina

Reviewer #2: **Yes: **Walaa Alkholy

---

## [Decision Letter · Decision Letter 1]

30 Jan 2023

Health system response to COVID-19 among primary health care units in Ethiopia: A 

qualitative study

PONE-D-22-03390R1

Dear Dr. Kassie,

We’re pleased to inform you that your manuscript has been judged scientifically suitable for publication and will be formally accepted for publication once it meets all outstanding technical requirements.

Kind regards,

Hubert Amu

Academic Editor

PLOS ONE

Additional Editor Comments (optional):

Reviewers' comments:

Reviewer's Responses to Questions

**Comments to the Author**

1. If the authors have adequately addressed your comments raised in a previous round of review and you feel that this manuscript is now acceptable for publication, you may indicate that here to bypass the “Comments to the Author” section, enter your conflict of interest statement in the “Confidential to Editor” section, and submit your "Accept" recommendation.

Reviewer #1: All comments have been addressed

Reviewer #2: All comments have been addressed

2. Is the manuscript technically sound, and do the data support the conclusions?

Reviewer #1: Yes

Reviewer #2: Yes

3. Has the statistical analysis been performed appropriately and rigorously? 

Reviewer #1: Yes

Reviewer #2: Yes

4. Have the authors made all data underlying the findings in their manuscript fully available?

Reviewer #1: Yes

Reviewer #2: Yes

5. Is the manuscript presented in an intelligible fashion and written in standard English?

Reviewer #1: Yes

Reviewer #2: No

6. Review Comments to the Author

Reviewer #1: The authors have addressed the comments that I recommended. I will recommend the manuscript for acceptance for publication.

Reviewer #2: Thanks for authors as all reply to my comments are accepted but I am interested to know authors reply to the comments of reviewer 2

7. PLOS authors have the option to publish the peer review history of their article (what does this mean?). If published, this will include your full peer review and any attached files.

Reviewer #1: **Yes: **Folorunso O. Fasina

Reviewer #2: **Yes: **Walaa Alkholy

---

## [Editor Report · Acceptance letter]

3 Feb 2023

PONE-D-22-03390R1 

Health system response to COVID-19 among primary health care units in Ethiopia: A 
qualitative study 

Dear Dr. Kassie:

I'm pleased to inform you that your manuscript has been deemed suitable for publication in PLOS ONE. Congratulations! Your manuscript is now with our production department. 

Kind regards, 

on behalf of

Dr. Hubert Amu 

Academic Editor

PLOS ONE